# Semi-field and surveillance data define the natural diapause timeline for *Culex pipiens* across the United States

Eleanor N. Field[1], John J. Shepard[2], Mark E. Clifton[3], Keith J. Price[4], Bryn J. Witmier[4], Kirk Johnson[5], Broox Boze[6], Charles Abadam[7], Gregory D. Ebel[8], Philip M. Armstrong[2], Christopher M. Barker[9] & Ryan C. Smith [1✉]

Reproductive diapause serves as biological mechanism for many insects, including the mosquito *Culex pipiens*, to overwinter in temperate climates. While *Cx. pipiens* diapause has been well-studied in the laboratory, the timing and environmental signals that promote diapause under natural conditions are less understood. In this study, we examine laboratory, semi-field, and mosquito surveillance data to define the approximate timeline and seasonal conditions that contribute to *Cx. pipiens* diapause across the United States. While confirming integral roles of temperature and photoperiod in diapause induction, we also demonstrate the influence of latitude, elevation, and mosquito population genetics in shaping *Cx. pipiens* diapause incidence across the country. Coinciding with the cessation of WNV activity, these data can have important implications for mosquito control, where targeted efforts prior to diapause induction can decrease mosquito populations and WNV overwintering to reduce mosquito-borne disease incidence the following season.

[1] Department of Entomology, Iowa State University, Ames, IA, USA. [2] Department of Environmental Sciences, Connecticut Agricultural Experiment Station, New Haven, CT, USA. [3] North Shore Mosquito Abatement District, Northfield, IL, USA. [4] Pennsylvania Department of Environmental Protection, Harrisburg, PA, USA. [5] Metropolitan Mosquito Control District, St. Paul, MN, USA. [6] Vector Disease Control International, Broomfield, CO, USA. [7] Suffolk Mosquito Control District, Suffolk, VA, USA. [8] Department of Microbiology, Immunology and Pathology, Colorado State University, Ft. Collins, CO, USA. [9] Department of Pathology, Microbiology and Immunology, School of Veterinary Medicine, University of California, Davis, Davis, CA, USA. ✉email: smithr@iastate.edu

Insects are one of the most diverse lifeforms on the planet, relying on a myriad of evolutionary adaptations to survive adverse ecological environments and climate conditions. This includes a state of dormancy known as diapause that is used to facilitate overwintering survival in temperate regions[1,2], which depending on species can occur during the egg, larval, or adult stages[3].

The northern house mosquito, *Culex pipiens*, is an important vector of mosquito-borne pathogens such as West Nile virus (WNV) and Saint Louis encephalitis virus (SLEV), and serves as an important model to understand diapause physiology in insects[3–5]. Evidence suggests that the immature life stages (larvae and pupae) respond to photoperiod and temperature cues to promote facultative diapause in adult female mosquitoes following eclosion[6–8], where the resulting females forego blood-feeding[9,10] and remain in an arrested reproductive state characterized by small primary ovarian follicles[3,11,12]. Additional physiological changes to the cuticle[13,14] and reduced diuresis[15] protect against desiccation, while alterations to host metabolism increase lipid and glycogen storage[16–20] to help sustain overwintering survival.

Photoperiod and temperature are long-established components of mosquito facultative diapause induction, where short-day lengths and cool temperatures promote this physiological state[6,8]. Previous studies with *Cx. pipiens* suggest that day lengths under 15 h can elicit the diapause under laboratory conditions using cool temperatures (18–22 °C)[6,8]. Evidence suggests that diapause induction is stronger with shorter photophases and lower temperatures, where ~100% of a colony can be induced under 12 h of daylight and at 18 °C[6]. However, higher temperatures can revert diapausing individuals or suppress diapause entry, even at lower photophases[5,7,21], demonstrating the combined importance of photoperiod and temperature in defining the diapause state.

Laboratory experiments to induce diapause using a short photoperiod (9:15) and cool temperature (19 °C) have produced significant insights into the hormonal regulation[4,18,22,23] and molecular physiology of *Cx. pipiens* diapause[17,24–26], yet these experimental conditions in the laboratory do not capture the natural fluctuations in daily temperature that encompass the realistic end-season conditions that promote diapause induction. Limited studies have addressed the diapause induction timeline in natural populations of *Cx. pipiens*. Semifield experiments in Ontario, Canada, recorded diapause incidence as early as July, with peak rates in mid-August[21], while field-collected mosquito samples in Boston, USA revealed low levels of diapause incidence beginning in mid-August, with peak incidence in late September/ early October[27]. In addition, evidence suggests that there is annual variation in diapause induction, with peak periods of induction varying by ~2 weeks between years[27]. Together, this geographic and yearly variability in diapause incidence highlight our limited understanding of *Cx. pipiens* diapause under natural conditions.

Herein, we perform laboratory and semifield studies to examine diapause induction in a laboratory population of *Cx. pipiens*, confirming the requirements of both photoperiod and temperature on diapause induction, as well as provide temporal evidence of the natural seasonal conditions that promote diapause in central Iowa, USA. To place these data in the context of diapause incidence in natural mosquito populations, we leverage gravid *Cx. pipiens* population data from Iowa and multiple locations across the United States to serve as a proxy for diapause incidence. These data suggest that temperature, latitude, elevation, and *Culex* population genetics significantly impact natural diapause ecology. Together, our data provide significant new insight into the complexity of *Cx. pipiens* diapause induction and its influence in end-of-season mosquito population trends. These results have important public health implications for mosquito-borne disease transmission, and increase our understanding of how a globally changing climate may extend mosquito activity and influence mosquito overwintering.

## Results

**Diapause induction requires both short-day lengths and cool temperatures**. With the intention to use a laboratory colony of *Culex pipiens* originally isolated from Ames, Iowa and maintained for ~16 years in the laboratory (without selection for the diapause state), we first wanted to demonstrate diapause induction in this *Cx. pipiens* population through laboratory experiments. To approach this question, we reared *Cx. pipiens* from first-instar larvae to adults under different laboratory conditions (Fig. 1a) to examine the influence of temperature alone (cold; 16:8 L:D, 19 °C), photoperiod alone (dark; 9:15 L:D, 25 °C), and the combined effects of temperature and photoperiod (diapause; 9:15 L:D, 19 °C) to promote adult diapause[10,18,28].

Using arrested ovarian development as a proxy for diapause induction[29], we examined ovarian follicle length in individual female adult mosquitoes 6–8 days post-eclosion[6,7] from each experimental condition (Fig. 1b). Interestingly, autogenous mosquitoes (displaying follicle maturation without a blood meal) were detected in all experimental conditions (Supplementary Fig. S1), suggesting that low rates of autogeny exist in our laboratory colony of *Cx. pipiens*. Across experimental conditions, average follicle size was comparable under the cold and dark conditions with standard laboratory rearing conditions (control), yet were significantly reduced under diapause conditions (Fig. 1b). Using a strict follicle size cutoff of ≤50 μm to designate individuals in diapause[29], only the cool temperatures and short photoperiod of the "diapause condition" produced individuals in the diapause state (Fig. 1b), as demonstrated by the arrested follicle morphology in diapausing individuals (Fig. 1c). This confirms that diapause induction in our laboratory colony of *Cx. pipiens* requires both short photoperiod and cool temperatures similar to the previous studies[6,11].

Additional previously defined physiological features of diapause[10,18,30] were also explored to further validate diapause induction in our laboratory colony. This includes increased lipid accumulation (Fig. 1d)[18,31], a larger body size (Fig. 1e)[30], and reduced blood-feeding behavior (Fig. 1f)[10], which together confirm the diapause state. Moreover, we demonstrate that our non-diapause rearing conditions can have additional influence on mosquito physiology, most notably the independent effects of temperature and short photoperiod on body size (Fig. 1e), and decreased feeding behavior under cool temperature conditions (Fig. 1f).

**Evaluating natural diapause induction in semifield experiments**. While laboratory studies are required to understand the physiological aspects of mosquito diapause, the environmental conditions used to promote diapause induction in the laboratory do not accurately depict the natural end-of-season conditions in temperate climates where temperatures are variable and photoperiods are less extreme than the 9-h photoperiod typically used in laboratory studies of mosquito diapause (Fig. 2a). In an effort to better understand the natural diapause conditions for *Cx. pipiens*, we performed a 2-year semifield study in Ames, Iowa, USA (Fig. 2b), enabling a structured approach to examine the life history conditions that result in diapause under natural conditions. To approach this question, we reared *Cx. pipiens* from our laboratory colony at semifield locations (Supplementary Fig. S2) from first-instar larvae to adults at different timepoints according to epidemiological week from July through late September

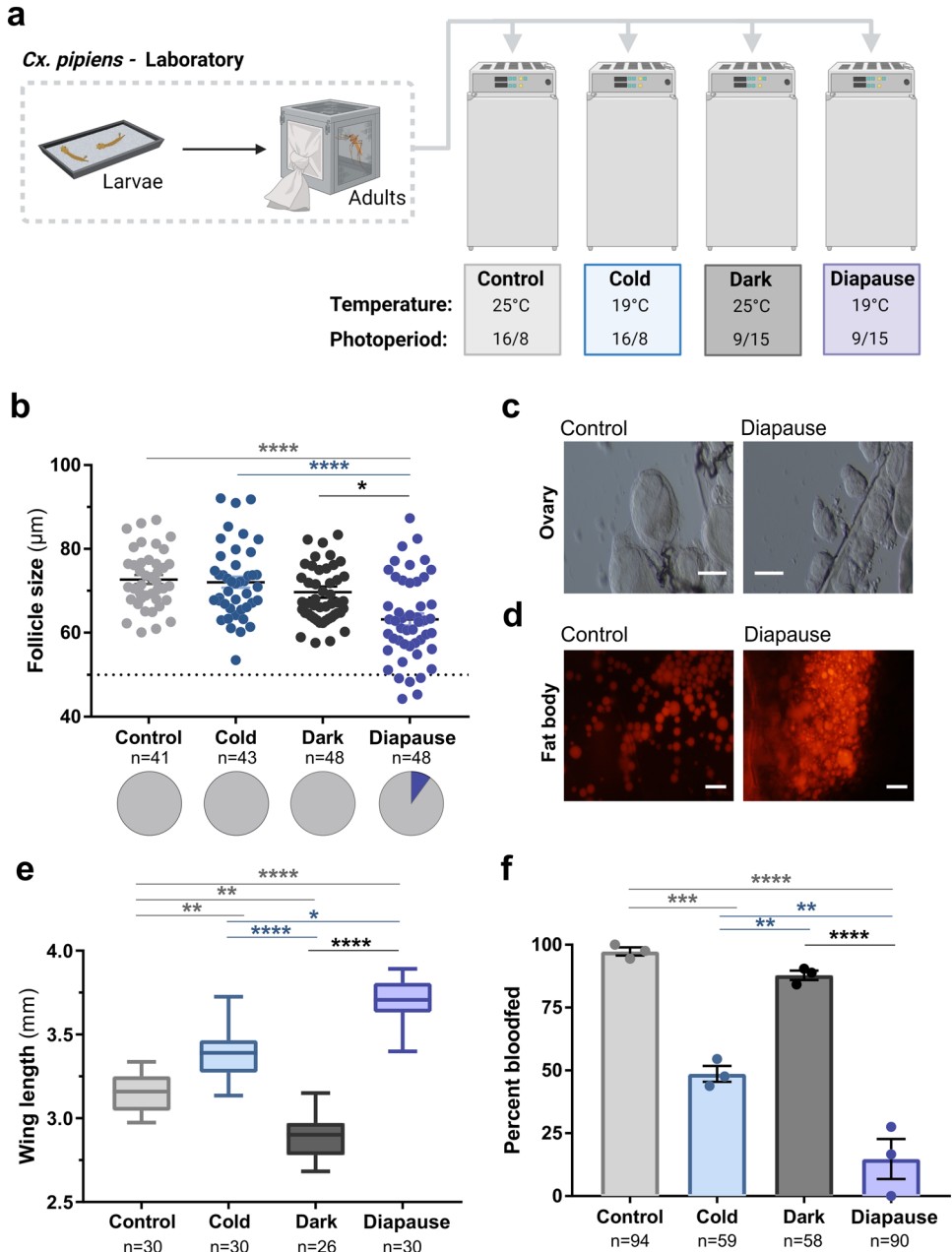

**Fig. 1 Laboratory conditions to explore Culex diapause induction.** *Culex pipiens* first-instar larvae were either maintained under standard insectary conditions (control) or transferred to rearing conditions to examine the effects of cold temperature (cold), short photoperiod (dark), or known conditions to promote diapause (**a**). Adult females were collected from these respective conditions 6–8 days post-eclosion for downstream experiments. **b** Ovary dissections were performed on adult mosquitoes from each rearing condition to determine the average primary follicle size to confirm reproductive diapause. The average primary follicle size is depicted for each individual mosquito, with an average follicle size was <50 μm (dotted line) used to confirm reproductive diapause. Solid black lines represent the median values for each experimental condition, while pie charts display the percentage of mosquitoes in diapause under each condition. Additional confirmations of mosquito diapause were performed by examining differences in ovarian follicle morphology (**c**) and fat body lipid staining with Nile Red (**d**) between mosquitoes reared under control or diapause conditions. Scale bars denote 50 μm in (**c**) and 20 μm in (**d**). **e** Wing-length measurements on adult females were performed as a proxy body size for each condition. **f** Blood-feeding behavior was evaluated in adult female mosquitoes from each treatment by challenging with an artificial membrane feeder containing sheep blood ($N = 3$). Significance was determined in follicle size and wing-length experiments using Kruskal–Wallis with a Dunn's post test. Blood-feeding experiments were analyzed using a one-way ANOVA test and Tukey post hoc comparisons to compare experimental treatments. Asterisks denote significance (*$P < 0.05$, ** $P < 0.01$, ***$P < 0.001$, ****$P < 0.0001$). n number of individual mosquitoes examined.

(Fig. 2b). Initially in 2020, groups of lab-reared first-instar *Cx. pipiens* larvae were placed outside every 3 weeks from week 30 to week 39 (July to late September; Fig. 2b), with each group representing an approximate 1-h difference in photoperiod (Supplementary Fig. S3) ranging from ~15 to 12 h of daylight at

the onset of larval development. A similar approach was employed in 2021, with experimental groups deployed at weeks 30, 33, 34, 37, 38, and 39 (Fig. 2b and Supplementary Fig. S3) to provide further resolution into the natural diapause induction timeline.

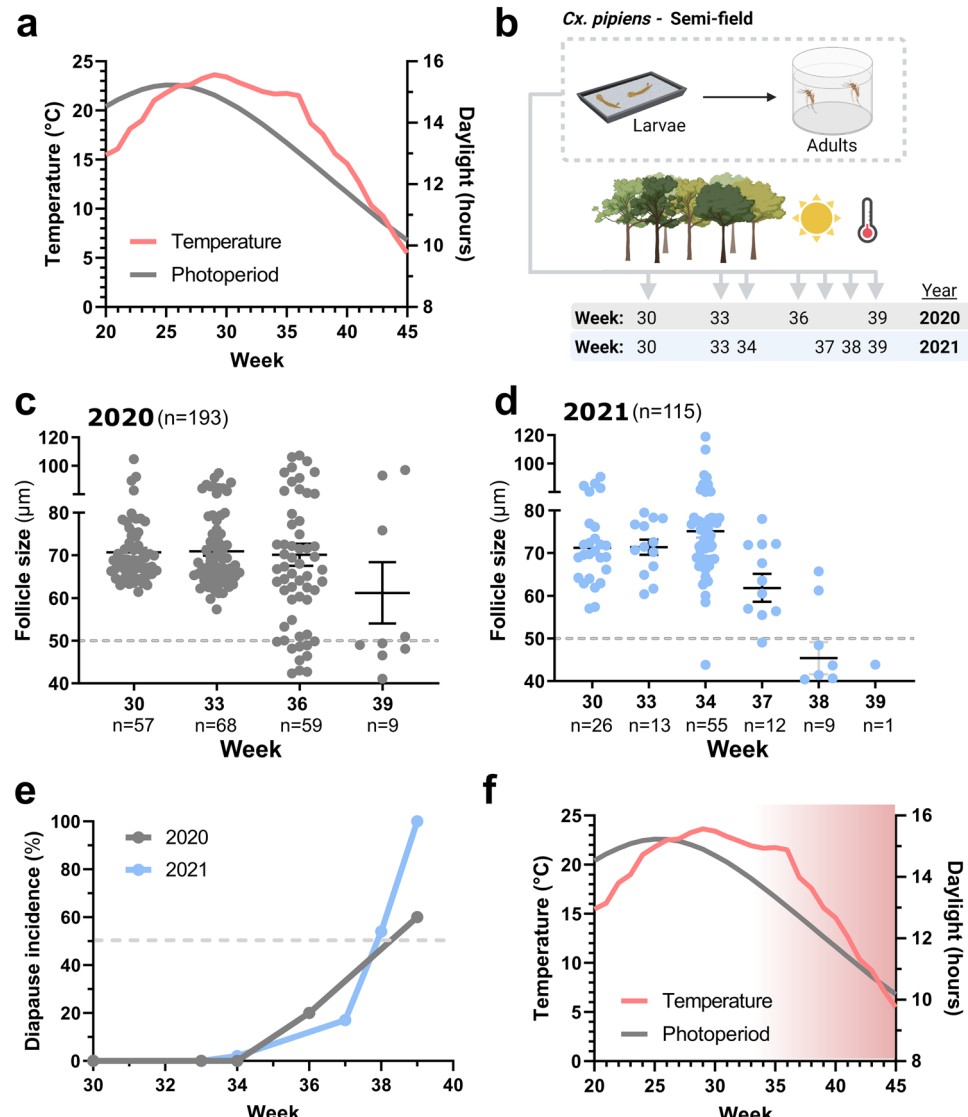

**Fig. 2 Examining the natural diapause timeline in semifield experiments. a** Temperature (°C) and photoperiod (daylight) averages (2009–2019) are displayed for the study site location (Ames, IA) over the mosquito season (weeks 20–45). **b** Overview of the semifield experiments performed in 2020 and 2021 where first-instar larvae from a lab colony of *Cx. pipiens* were placed outside at weekly timepoints (between week 30 and 39). Larval groups were reared in a semifield environment, with resulting adult females used for downstream experiments 6–8 days post-eclosion. A total of 193 adult female mosquitoes were collected in 2020 (**c**) and 115 adult females from 2021 (**d**) to determine if mosquitoes were in reproductive diapause. For **c** and **d**, each dot represents the average follicle size for an individual mosquito, with a 50 μm threshold (dotted line) used to determine individuals in the diapause state. The mean follicle size (+/− SEM) is displayed for each experimental cohort. n number of individual mosquitoes examined. **e** The percentage of adult diapause incidence recorded by larval onset week is summarized for 2020 and 2021. The dashed line corresponds to when 50% of the mosquitoes in a given larval onset week result in reproductive diapause. From our experiments, we define an approximate period of diapause receptivity (**f**), where larval development under temperature and photoperiod conditions of 13.5 h of daylight and average temperatures under 20 °C may give rise to adult reproductive arrest, resulting in increasing diapause incidence (red gradient) as these environmental conditions continue to decrease over the course of the season.

To determine which mosquitoes had entered diapause, we examined ovarian follicle size as in Fig. 1. In 2020, a total of 193 adult female mosquitoes were examined from four experimental groups, of which diapause was detected in mosquitoes with a rearing onset initiated during weeks 36 and 39 (early- and late September; Fig. 2c and Supplementary Table S1). Similar results were obtained in 2021, where 115 mosquitoes across six experimental groups were examined, with diapause detected in groups with a rearing onset as early as week 34 (late August) and increasing in intensity through the remainder of the experimental timepoints initiated in September (Fig. 2d and Supplementary Table S1). For both years, diapause induction was strongest (≥50% of mosquitoes in diapause) after week 38 (late September;

Fig. 2d), when immature mosquito development occurred with ~12 h of daylight and an average temperature of 15 °C (Fig. 2f), yet was readily detected in groups reared under 13.5 h of daylight and ~20 °C (Fig. 2f).

To further examine this receptive period able to promote diapause induction, we placed lab-reared pupae under semifield conditions each week from weeks 36–40 (September to early October) and evaluated their ability undergo reproductive diapause. While the transfer of lab-reared pupae to artificial diapause conditions (9:15 L:D, 19 °C) results in nearly 100% diapause induction[6], under our semifield conditions, diapause induction was inefficient, with diapause only detected in low frequency (6–17%) from pupae placed outside in weeks 39 and 40

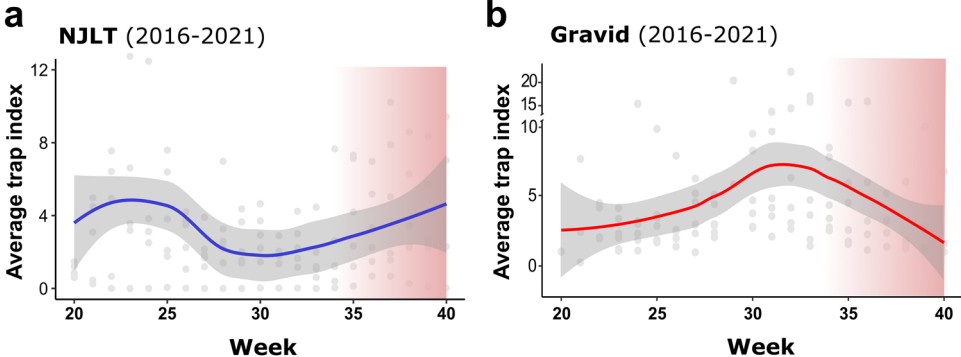

**Fig. 3 Iowa mosquito surveillance data confirm the natural diapause timeline.** Mosquito surveillance from central Iowa display mosquito populations trends that examine general population abundance (**a**) or represent only reproductive populations (**b**). Data in (**a**) represent *Cx. pipiens* group abundance measured using New Jersey Light Traps (NJLT) from 14 sites in central Iowa, while data in (**b**) display Gravid trap data from 16 sites in central Iowa that monitor reproductive female *Cx. pipiens* populations. For both **a** and **b**, collections were performed from 2016–2021 from weeks 20 to 40. Individual gray dots represent yearly trap index averages by week, with the blue (NJLT) or red (Gravid) lines representing the loess-smoothed mean and 95% confidence intervals (gray-shaded area). The approximate period of diapause receptivity (as defined by our semifield experiments) is displayed by the red gradient.

(Supplementary Fig. S4).This corresponds to the approximate environmental conditions in mid-to late September that result in ≥50% diapause induction in our larval groups (Fig. 2e), suggesting that the environmental signals during late September may be driving diapause induction in our semifield conditions. In addition, the low frequency of diapause induction at the pupal stages is suggestive that cumulative exposure over immature developmental stages may enhance diapause induction as previously suggested[7].

Based on our results (Fig. 1) and other previous studies[6,11], the temperature is an important, yet complex variable in the context of diapause induction. When examined between years, there is a slight variation in diapause induction (Fig. 2e), likely the result of temperature differences shifting the timing of diapause induction between years (Supplementary Figs. S3 and S5 and Supplementary Table S1). In addition, our data highlight the potential importance of diurnal fluctuations in temperature that occur during periods of diapause receptivity (Supplementary Fig. S6), where daily low temperatures may be able to sustain diapause induction signals even during the exposure to high daily temperatures (>30 °C) in our semifield studies that would typically "break" diapause induction[6]. Temperature also had a significant influence on mosquito development and survival as temperatures declined in late summer/early fall. As the season progressed, our experimental groups displayed increased developmental times, most notably slowing larval and pupal development, ultimately influencing adult eclosion (Supplementary Fig. S3). Moreover, in both years of our semifield study, larvae placed outside in late September (weeks 38 and 39) experienced significant mortality in immature stages due to suboptimal temperatures for mosquito development and survival, resulting in the low numbers of individual mosquitoes that contributed to our analysis of reproductive diapause (Fig. 2c). However, these impacts on survival were based only on observations and were not directly measured.

**Mosquito surveillance data inform diapause induction in Iowa field populations.** While our semifield study provides valuable new insight into the conditions and timing of natural diapause induction (Fig. 2), these experiments were performed using a laboratory-derived population of *Cx. pipiens* and may not fully capture diapause induction, such that natural field populations of mosquitoes may be more receptive to photoperiod and temperature. To measure the natural diapause timeline in populations of *Cx. pipiens*, we utilized long-term mosquito surveillance

data from central Iowa to capture adult *Cx. pipiens* population trends in natural field settings.

Using different trap types to estimate general mosquito abundance (New Jersey light trap; NJLT) or blood-fed/reproductive mosquito populations (grass infusion-baited gravid traps; gravid), we examined Cx. pipiens population dynamics in central Iowa (Supplementary Fig. S7) from 2016 to 2021 as a proxy for diapause induction in natural field populations. NJLT data demonstrate early-season peaks (May/June) in *Culex pipiens* group[32–34] abundance, which taper mid-summer (July) before a late-season rise in September (Fig. 3a). In contrast, gravid *Cx. pipiens* populations peak mid-summer (week 32), then decline by 61% by week 40 (Fig. 3b). Comparisons of NJLT and gravid population trends using a linear regression of annual slope values over weeks 30–40 were significant ($P = 0.0079$), supporting that only reproductive gravid *Cx. pipiens* populations are declining during the late summer (Supplementary Fig. S8). When placed in the context of diapause induction established in our semifield experiments (Fig. 2), gravid adult field populations experience significant declines during September, corresponding with the approximate environmental signals able to promote diapause in our semifield studies. Moreover, the cessation of mosquito surveillance after week 40 (October) tightly corresponds with the high rates of diapause in our semifield groups suggesting that after the beginning of October most emerging females will be in diapause (Fig. 3b).

***Cx. pipiens* diapause incidence shares similar timelines across the United States.** Based on our observations of *Cx. pipiens* diapause induction in semifield (Fig. 2) and natural field conditions in Iowa (Fig. 3), we wanted to similarly examine potential diapause timelines across the United States. To approach this question, mosquito surveillance data was collected from across the country (California, Colorado, Connecticut, Illinois, Minnesota, Pennsylvania, and Virginia) to examine adult *Cx. pipiens* population dynamics, similar to that described in Iowa (Fig. 3).

For locations with both NJLT and gravid trap types, linear regressions of annual slopes by trap type confirm that gravid trends were distinct from general population trends (NJLT) from weeks 30 to 40 at each location (Iowa, $P < 0.01$; California, $P < 0.001$; Colorado, $P < 0.05$; Connecticut, $P < 0.0001$) (Supplementary Fig. S8). When historical gravid trapping data were used to compare end-season (weeks 30–40) across the country, there was a consistent decline in *Cx. pipiens* gravid populations from mid-summer (July, week 30) to late summer/early fall across the

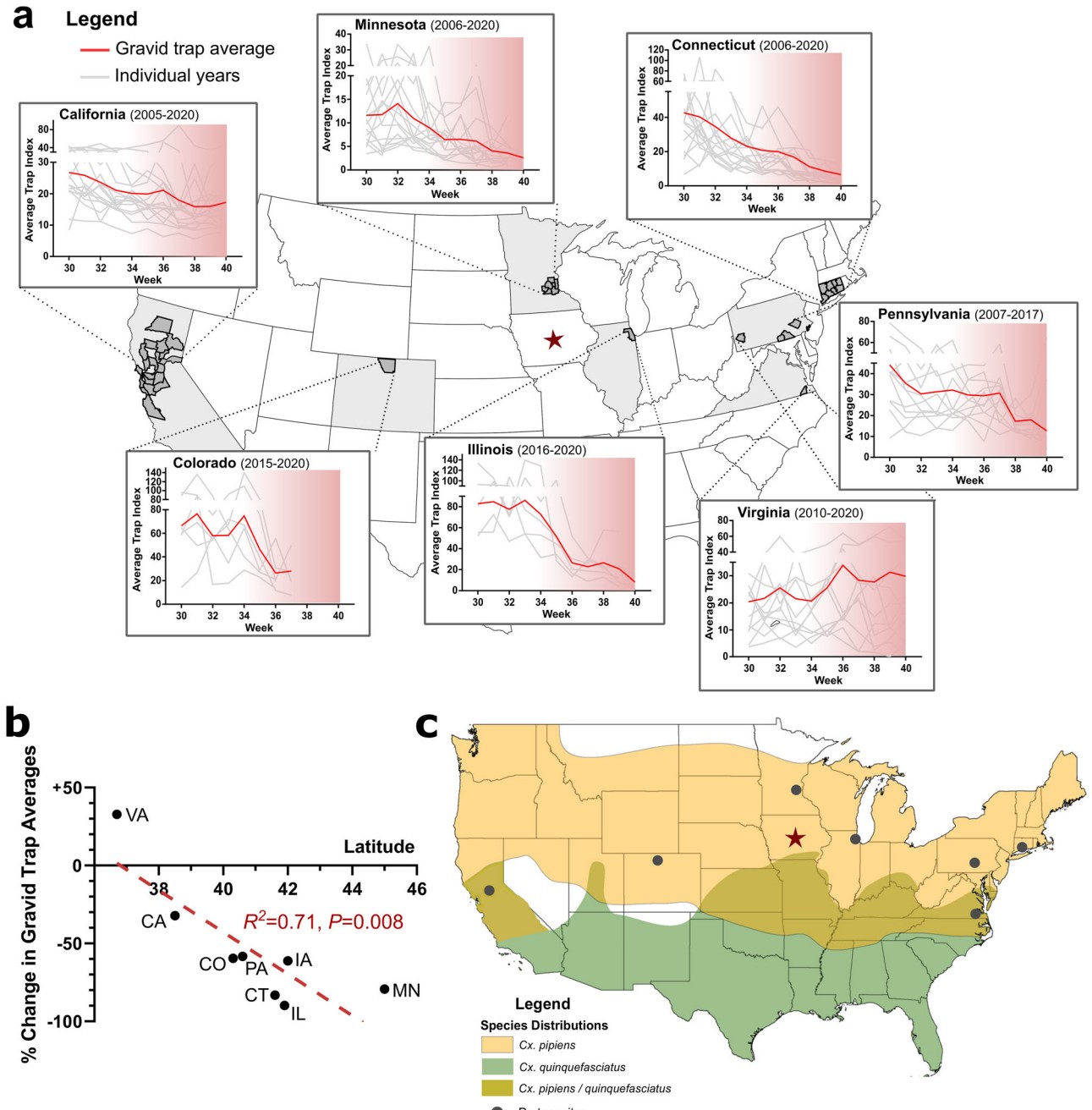

**Fig. 4 Gravid population trends from across the United States provide insight into the biotic and abiotic factors that influence *Cx. pipiens* diapause.**
**a** End-of-season *Cx. pipiens* gravid trap data (weeks 30–40) provided by locations across the United States. States included in our analysis are denoted in light gray, with individual county-level data included in the statewide analysis shown in dark gray. Iowa is denoted by the red star. For each state, red lines represent the loess-smoothed mean of the gravid (reproductive) population abundance, while gray lines display data from an individual year included in our analysis. The approximate period of diapause receptivity (as defined by our semi-filed experiments) is displayed by the red gradient. **b** Sites at higher latitudes demonstrated more pronounced declines in gravid trap averages from weeks 30–33 to week 40. **c** Map of the reported distributions of *Cx. pipiens* (yellow), *Cx. quinquefasciatus* (green), and areas of potential genetic hybridization between these species (olive). Site locations providing gravid trap data are shown by the gray dots, while Iowa is denoted by the red star.

United States (Fig. 4a), with the exception of gravid populations from Suffolk, Virginia which displayed a slightly increased trend in *Cx. pipiens* abundance (Fig. 4a). These observations coincide with the decline of gravid populations in Iowa over August and September (Fig. 3), where gravid populations in most location undergo notable declines in gravid *Cx. pipiens* populations from week 30 to 40 ranging from ~32 to 90% (Fig. 4b).

Lower latitude sites (Suffolk, Virginia, and California) (Supplementary Fig. S9) had less pronounced population declines

(and even increased *Cx. pipiens* gravid populations; Fig. 4b), highlighting the significant influence ($R^2 = 0.69$, $P = 0.01$) of latitude on end-of-season *Cx. pipiens* dynamics (Fig. 4b). This coincides with similar effects of latitude and photoperiod on diapause induction in other insects[35,36] and mosquito species[37,38]. Moreover, in both Virginia and California, evidence suggests that these regions are within hybridization zones between *Cx. pipiens* and *Culex quinquefasciatus*[39–41] (Fig. 4c), a morphologically identical species that does not undergo

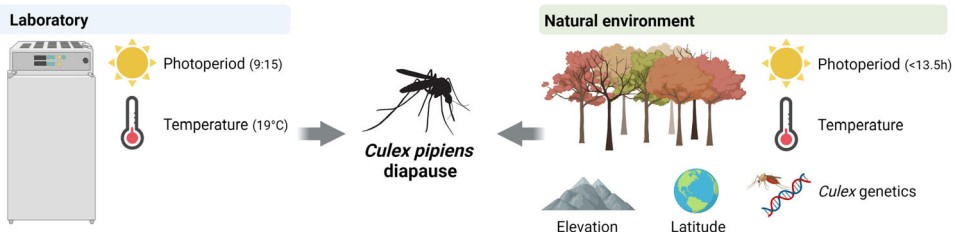

**Fig. 5 Overview of the factors that influence *Cx. pipiens* diapause in the laboratory and under natural conditions.** The combination of short-day lengths and low temperatures form the basis for diapause induction in laboratory studies of *Cx. pipiens*, yet additional ecological factors (latitude, elevation, population genetics) influence diapause dynamics in natural settings.

diapause[8,30,42]. As a result, the absence or low incidence of diapause in these locations may be due to the respective misidentification of *Cx. quinquefasciatus* as *Cx. pipiens*, the hybridization of these species resulting in intermediate diapause phenotypes[8,30], or the potential that these could be populations of *Cx. pipiens* f. molestus that do not undergo reproductive diapause[43].

To make comparisons of diapause-relevant environmental factors across sites, average weekly day length and average temperature values were compiled from each location (or comparably close locations) (Supplementary Fig. S10). Temperatures across all study sites noticeably declined over the 10-week period, with Minnesota (Minneapolis/St. Paul) and Colorado (Larimer County) having the coldest average temperature (~10 °C) at week 40 (Supplementary Fig. S9). Connecticut, Illinois (Chicago), and Pennsylvania displayed comparable average temperatures (~15 °C) at week 40 (Supplementary Fig. S10), similar to the average temperatures in Iowa at this timepoint (Fig. 2f). Of note, the two locations (Northern California; Suffolk, VA) with the smallest changes in gravid trap numbers (Fig. 4b), which suggest little to no diapause, had much higher temperatures across the entire 10-week period, with week 40 averages of ~20 °C (Supplementary Fig. S10). Although average day lengths varied slightly at the start (week 30), with the highest latitude location (Minnesota) having slightly longer day lengths, all included locations converged to near identical levels (~12 h) by the fall equinox (week ~38).

Our nationwide gravid trap data also allude to the influence of elevation in *Cx. pipiens* diapause induction, where surveillance data from Colorado display sharp declines in gravid populations beginning in late August (week 34) and the termination of surveillance activities only weeks later (week 37; Fig. 4a). This accelerated timeline suggests that the high altitude of Larimer County, CO (Supplementary Fig. S9) enhanced the environmental signals that promote diapause induction, similar to that described in other mosquito species[37,44].

## Discussion

Although diapause is a critical component to the success of *Culex pipiens* overwintering survival in temperate regions, our understanding of the environmental signals that promote diapause induction under natural conditions has thus far been limited. Using laboratory and semifield experiments to inform historical mosquito surveillance trends from across the county, we provide a definitive diapause induction timeline broadly shared in true *Cx. pipiens* populations across the United States. While temperature and photoperiod are integral components of diapause induction in laboratory and field conditions, our data suggest that latitude and elevation can further amplify the effects of temperature and photoperiod under natural conditions, as well as

highlight the importance of the influence *Culex* population genetics in defining diapause incidence (Fig. 5).

There are many physiological and behavioral changes associated with diapause[10,11,13,31,45], yet the key feature to confirm reproductive diapause in *Cx. pipiens* is an arrested ovarian development phenotype[11,46,47]. Morphologically this corresponds to arrested ovarian follicle development, in which follicles lack yolk granulation and become stunted in size[48]. Arrested ovarian development has also been quantified using either direct measurements of the primary follicle[5,18,25,29,30], or the size ratio of the primary follicle to the secondary follicle[6,42,46,49] to establish diapause. In our experiments, we relied on measurements of the primary follicle to determine reproductive diapause. Previous experiments have defined diapause using this methodology with follicle lengths ranging from 30 to 70 μm[5,18,25,29,30,49]. However, our *Cx. pipiens* colony produced primary follicle lengths of ~75 μm under standard rearing conditions. As a result, a more conservative cutoff of ≤50 μm was used to confirm reproductive diapause in our lab colony at the risk of potentially excluding some individual mosquitoes that did not meet these strict criteria in our laboratory and semifield experiments.

In laboratory experiments using an established colony of *Cx. pipiens*, we observed a relatively low frequency of diapause induction when we applied our strict criteria for ovarian arrest. Although diapause is genetically determined (reviewed in ref. [48]), our colony of *Cx. pipiens* has been maintained in artificial laboratory conditions after its initial colonization (~2005) without re-invigoration from wild-caught specimens or artificial diapause cycling[29]. As a result, our laboratory colony may have become desensitized to the thermal and photoperiod cues required to promote diapause, similar to the influence of artificial rearing conditions on diapause incidence in other insect species[50–52]. In addition, our laboratory diapause induction experiments used a fluorescent light source, which previous studies have suggested is less efficient than incandescent light at promoting diapause in *Cx. pipiens*[8]. While at present we cannot provide a clear explanation for the low rates of diapause induction in our laboratory experiments, the same lab colony was used in our semifield experiments where diapause incidence reached in excess of 50%, suggesting that the predisposition for diapause remains in our *Cx. pipiens* colony. Factors such as the length of crepuscular periods, more extreme low daily temperatures, or diurnal temperature fluctuations may be important natural variables that can overcome diapause de-sensitivity resulting from long-term colonization.

While short photoperiod (9:15 L:D) and cool temperatures (19 °C) are traditionally used to initiate diapause in the laboratory[26,28,53], these experimental conditions do not accurately reflect the natural onset of diapause in the field. Photoperiod values only vaguely provide a timeline of diapause potential between the summer solstice (15 h of light) where low-

level diapause is possible[6,21] and the winter solstice (9 h of light) where temperatures prove impossible for mosquito survival and development in temperate climates. Similarly, as temperatures can subvert diapause induction[6], the potential for natural fluctuations in the end-season may affect presumed diapause timelines derived from stable lab combinations of photoperiod and temperature. However, only a limited number of studies have examined *Cx. pipiens* diapause in field or semifield settings[21,27,49].

Through our semifield studies conducted over a 2-year period (2020–2021), we systematically reared mosquitoes over weeks 30–40 to capture the natural conditions that promote diapause induction in *Cx. pipiens*. From these experiments, we define a timeline of brood receptivity to diapause beginning in late August (week 34) when immature mosquitoes have the potential to emerge in adult diapause. With increasing diapause incidence as the season progresses, our data support that there is a critical field photoperiod in mid-September (week 38) where ~50% of the immature *Cx. pipiens* population reared at this time emerged in the adult diapause state. This closely coincides with the results of a previous semifield study in Boston, USA, where peak diapause incidence occurred in late September and early October[27]. While diapause can be induced in pupae under artificial diapause conditions[6], the low occurrence of pupal diapause in our semifield study during these peak periods of diapause induction suggests that cumulative environmental signals experienced over all immature stages may enhance diapause incidence as previously proposed[7].

While our semifield experiments are informative in exploring natural diapause induction, factors such as the gaps between experimental cohorts and the reliance on a laboratory colony of *Cx. pipiens* are known limitations. To overcome this, we employed the use of *Cx. pipiens* surveillance data as a proxy for adult diapause incidence[21]. Through the use of a multi-year dataset that captured long-term weekly adult abundance, we demonstrate that gravid female mosquito populations begin to decline in August until they are effectively depleted by October. Although some of these trends will be in part from encroaching cold temperatures that reduce mosquito abundance[54] and blood-feeding behavior (as demonstrated in our laboratory experiments), these observations closely align with the natural diapause conditions defined in our semifield study. Initially focused on surveillance data from Iowa, additional data from across the country provided an opportunity to study diapause timelines in a larger context, where surveillance data at the national scale revealed that latitude, elevation, and *Culex* population genetics may influence *Culex* diapause incidence in addition to temperature and photoperiod.

When our data are placed in the context of the wide geographic range within the United States, the importance of latitudinal patterns on diapause incidence begin to emerge for *Cx. pipiens*, where we identify a gradient in the effects of latitude and elevation on diapause incidence, similar to that described in other mosquito species[37,44,55,56]. Moreover, our data suggest that *Culex* population genetics may also have significant influence on diapause induction where *Cx. pipiens* hybridization with the morphologically indistinguishable *Cx. quinquefasciatus* may result in intermediate diapause phenotypes[30]. Together with the potential of non-diapausing *Cx. pipiens* f. *molestus* populations, the genetics of local *Culex pipiens* s.l. populations may contribute to the absence or low incidence of diapause in a given location, where only genetically pure populations of *Cx. pipiens pipiens* enter reproductive diapause[6,49]. However, only limited studies have examined *Cx. pipiens* s.l. population genetics in the United States[39,57–63], highlighting the important need to better define these important vector species. Of note, few pure *Cx. pipiens pipiens* were detected in northern California[58,64], which may

account for the weak population declines observed in the gravid trap data from California. This coincides with previous observations in northern California that *Cx. pipiens* s.l. do not enter diapause[49]. The population genetics of *Cx. pipiens* s.l. in Virginia have not been previously examined, yet due to the geographic location, there is likely hybridization within the *Cx. pipiens* complex that may similarly partially account for the observations in the gravid trap data.

Although temperature is considered an important signal for diapause induction, its contributions to diapause have primarily been evaluated under stable conditions in the laboratory, with little insight into weekly temperature variability and daily temperature fluctuations that occur in nature. As a result, identifying periods of diapause receptivity to temperature alone is difficult, especially when temperature can influence *Culex* species immature development times[65,66], as well as adult survival, blood-feeding, fecundity, and abundance[54,66]. Similar to these experiments, we observed extended larval development times with cooler average temperatures, which may allow for the increased accumulation of lipids in subsequent diapausing adults[18]. However, as temperatures continue to drop in late summer and early fall when developing larvae can emerge in diapause, there is a tight balance between accumulating enough resources to enter diapause and not being able to survive temperatures that do not allow for further development. As such, temperature can be a highly confounding variable, one that likely accounts for small variations in the timing of diapause induction between years.

An additional, often overlooked aspect of temperature is the influence of daily temperature fluctuations on mosquito physiology. In *Aedes aegypti*, large diurnal temperature ranges negatively impact mosquito development[67], adult female fecundity[67], and vector competence[68]. Given that diurnal temperature ranges are largest in temperate climates during the late summer and early fall when mosquitoes are receptive to diapause, we hypothesize that these daily temperature fluctuations may similarly influence *Culex* physiology and diapause induction. Moreover, daily temperature fluctuations may protect the commitment to diapause if diapausing adults are exposed to higher temperatures that would regularly break reproductive diapause under laboratory conditions[5].

Although long inferred, the relationship of diapause induction to the cessation of West Nile virus (WNV) activity has yet to be fully explored. Our data provide strong support that the diapause incidence in the late summer/early fall coincides with the dramatic decline of human WNV cases and mosquito infection rates in late September and early October in temperate regions of the United States[32,69,70]. Moreover, since diapausing females do not blood feed[10], it argues that the overwintering of WNV in *Cx. pipiens*[71–74] occurs via vertical transmission as previously suggested[75,76]. Coincidently, the period of diapause receptivity in the late summer and early fall also corresponds the peak of WNV mosquito infection rates[32]. Although vertical transmission is an inefficient process[71–74], the increased prevalence of WNV infection in mosquito populations during the approximate time when conditions are favorable for diapause induction may enhance WNV overwintering in diapausing female mosquitoes. As a result, measures to control *Culex* populations prior to diapause induction may not only reduce mosquito populations in the following spring as previously suggested[77], but also limit WNV overwintering and subsequent disease burdens in the following season.

In summary, our findings provide a definitive examination of diapause induction in *Cx. pipiens* supported by laboratory, semifield, and field-collected surveillance data from across the United States. We demonstrate the dynamic nature of diapause ecology influenced by yearly variation in temperature, as well as

the effects of latitude, elevation, and mosquito population genetics that ultimately determine the overall end-season population structure of *Cx. pipiens* and its role in WNV transmission. Taken together, these data demonstrate the importance of mosquito diapause in defining periods of mosquito-borne disease transmission in the United States. With evidence suggesting that rising global temperatures can alter diapause incidence[78], the effects of climate change may extend transmission seasons and increase the incidence of mosquito-borne disease in temperate regions throughout the world[78–80].

## Methods

**Mosquito rearing**. A laboratory colony of *Cx. pipiens* mosquitoes originally isolated from field collections in Ames, Iowa has been constantly maintained in Iowa State University's Insectary at 25 °C, 85% RH, and 16:8 (L:D) on 10% sucrose ad libitum since ~2005. Larvae were fed using a 50/50 mix of crushed Milk-Bone® and Tetramin® fish food, while commercial sheep blood (Hemostat Laboratories) was used for egg production.

**Laboratory diapause induction experiments**. Laboratory experiments to examine diapause induction were performed by placing newly hatched first-instar mosquitoes in Percival incubators where they were reared under different experimental conditions (*Control*: 25 °C, 16:8 (L:D); *Cold*: 19 °C, 16:8 (L:D); *Dark*: 25 °C, 9:15 (L:D); *Diapause*: 19 °C, 9:15 (L:D)) to examine the independent and combined influence of temperature and photoperiod on diapause induction. Experimental conditions were selected based on the previous studies[10,26,30] in which diapause was induced by a short photoperiod (9:15) and cool temperatures (19 °C). Data from all experiments were collected from three or more independent biological replicates.

**Confirmation of reproductive diapause**. Ovaries were dissected from females aged 6–8 days in 1% PBS solution and mounted with Aqua-Poly/Mount (Polysciences Inc). To confirm reproductive diapause, primary follicle lengths were measured under ×200 magnification using an Olympus BX40 compound microscope according to ovary morphology as previously defined[28,29]. Ten measurements per ovary were recorded to calculate an average follicle length per individual. Individuals with average follicle size under 50 μm were defined as being in diapause[29].

**Wing-length measurements**. Measurements of wing length served as a proxy for mosquito body size[81,82], where the right wing was dissected and measured from the alula to the most distal tip of the wing under ×10 magnification using a dissecting microscope and Nikon imaging software (NIS Elements D 3.2).

**Blood-feeding behavior**. Approximately 20–30 adult females (6–8 days post-eclosion) from each experimental rearing condition, as described above, were challenged with defibrinated sheep blood (Hemostat Laboratories) using an artificial membrane system. After approximately 1 h, the number of mosquitoes with a visible blood meal were recorded (of the total) to calculate the percentage of mosquitoes taking a blood meal. Experiments were performed in three independent biological experiments.

**Lipid staining**. Nile Red (Thermo Fisher Scientific) was used to visualize differences in lipid stores of adult females (6–9 days old) reared under diapause (19 °C, 9:15 L:D) and control conditions (25 °C, 16:8 L:D). Fat bodies were dissected in 4% paraformaldehyde and stained using a 1:100 PBS dilution of 500 μg/ml Nile Red powder in acetone stock as previously described[18,31]. After incubating tissues for 10 min, samples were imaged using a Nikon 50i fluorescent microscope and processed with Nikon imaging software (NIS Elements D 3.2).

**Semifield studies of diapause induction**. To examine diapause induction in a natural setting, first-instar larvae from our laboratory colony were concurrently placed at three locations in Ames, Iowa, in 2020 (three replicates) and two locations in 2021 (two replicates, summarized in Supplementary Fig. S2) to mimic the emergence of mosquito populations at different timepoints throughout the season. In 2020, batches were placed outside every 3 weeks according to epidemiological week, from week 30 (July 19th) to week 39 (September 20th), approximately corresponding to a 1-h loss in daylight between each group. In 2021, batches were again placed outside at weeks 30, 33, and 39 to replicate experiments from the previous year. Unfortunately, larvae were not available to repeat the week 36 timepoint in 2021, however additional groups were included from weeks 34, 37, and 38 to provide additional resolution to diapause induction. Larval density in each batch was ~300–400 per tray in 1 L of distilled water. Mosquitoes were fed daily with 50 mg of a 50/50 mixture of Milk-Bone® mix and Tetramin® fish food. Upon pupation, pupae were placed into mosquito breeder eclosion chambers

(BioQuip), with adults were provided with 10% sucrose ad libitum for 6–8 days before collections to determine diapause incidence. Samples from each site location (replicates) were pooled for each experimental cohort (week) to examine follicle size and diapause induction. In addition, lab-reared pupae were placed outside in mosquito breeder eclosion chambers in 2021 from weeks 37 to 40 to compare diapause induction rates between mosquito life stages.

**Mosquito population trends in Iowa**. Mosquito surveillance was performed in two central Iowa counties (Polk, Story) by Iowa State University personnel or local public health partners from mid-May (week 20) through the first week of October (week 40). Mosquito collections were performed using infusion-baited Frommer Updraft Gravid Traps (John W. Hock Company) targeting gravid adult female mosquitoes at 16 sites over a six-year period (2016–2021; Supplementary Fig. S6). In addition, a total of 14 New Jersey light traps (NJLTs) were used to measure mosquito abundance (2016–2021). Trapping sites (gravid or NJLT) with less than three years of mosquito data were excluded from the study. All mosquito samples were identified using morphological keys[40,83] where possible. Due to damage to morphological features that help to define *Culex* species[84], mosquito identifications of *Culex pipiens* and *Culex restuans* from NJLT traps were defined as *Culex pipiens* group as previously[32,34]. Gravid trap specimens were identified to species (*Culex pipiens*)[83]. To normalize trapping efforts, raw mosquito counts were converted to a trap index (defined as the number of mosquitoes collected/number of trapping nights) and then averaged by week.

**Iowa climate and photoperiod data**. Daily temperature data (°C) was collected and averaged into weekly values by year using the Iowa Environmental Mesonet (https://mesonet.agron.iastate.edu) for the Southeast Ames station (IA0203). Photoperiod data for the duration of the study period was collected as hours of daylight for Des Moines, Iowa from daylight tables provided from an online Sunrise and Sunset table (www.timeanddate.com).

*National trends in diapause induction, temperature, and Culex species hybridization*. Mosquito surveillance data were provided from additional locations across the United States (California; https://vectorsurv.org/, Colorado, Connecticut, Illinois, Minnesota, Pennsylvania, and Virginia), representing the temperate range of *Cx. pipiens* in the United States[40]. The years included from each dataset are as follows: California (2005–2020), Colorado (2015–2020), Connecticut (2006–2020), Illinois (2016–2020), Minnesota (2006–2020), Pennsylvania (2007–2017), and Virginia (2010–2020). To reflect the end-of-season population trends, data was trimmed to reflect weeks 30–40 where applicable (Colorado ends at week 37, Chicago ended most weeks at 39), with raw mosquito counts normalized using trap index averages to account for differences in trapping efforts (number of mosquitoes/numbers of trapping nights) as previously performed for the Iowa dataset. Data from Illinois and Minnesota represent *Culex pipiens* group (a combination of the morphologically similar *Culex restuans* and *Cx. pipiens* species)[32,33], while records from other states were morphologically identified as *Cx. pipiens*.

Daily high- and low temperatures for all site locations were compiled using the Iowa Environmental Mesonet (https://mesonet.agron.iastate.edu). All locations reflect 10-year averages (2010–2020) where environmental data were paired to trapping locations as follows: California (Sacramento, CATSAC), Connecticut (statewide average, CT0000), Colorado (east Fort Collins, CO3006), Illinois (Chicago O'Hare International airport, ILTORD), Minnesota (Minneapolis-St. Paul, MNTMSP), Pennsylvania (southeastern PA, PAC003), and Virginia (Suffolk, VA8192). Photoperiod data was collected from an online Sunrise and Sunset table (www.timeanddate.com) for field locations in California (Sacramento), Colorado (Fort Collins), Connecticut (New Haven), Illinois (Chicago), Minnesota (Minneapolis), Pennsylvania (Philadelphia), and Virginia (Chesapeake) for 2021.

Elevation data for all site locations are provided as a county-level average value collected using https://en-gb.topographic-map.com/.

**Graphical mapping**. The topographical map of the United States highlighting counties with contributing mosquito abundance data was generated in R version 4.1.3 (R Development Core Team) using the "usmap" and "ggplot" packages.

The proposed ranges of *Cx. pipiens* and *Culex quinquefasciatus*, as well as proposed areas of hybridization, were created from a base map of the United States obtained in ArcGIS (https://www.arcgis.com/home/index.html), with mosquito ranges adapted from Darsie and Ward[40] and illustrated using Inkscape (https://inkscape.org/).

**Statistics and reproducibility**. Laboratory comparisons comparing ovarian follicle size and wing length were analyzed using Kruskal–Wallis with a Dunn's post test, while the percentage of blood-feeding across experimental groups was examined using a one-way ANOVA with a Tukey post hoc analysis. All statistical analyses were performed using GraphPad Prism 7.0. Average weekly temperature data were visualized using loess smoothing in R (version 3.6.3). Gravid and NJLT population trends for weeks 30–40 were examined for the Iowa dataset and other locations where NJLT data were provided using yearly slope values and negative binomial regressions with an unpaired *t* test to demonstrate significant differences in the dynamics of gravid populations. Mosquito population trends were also evaluated

by calculating the difference in the average trap index at weeks 30–33 and week 40, and displaying these trends as a percent change ((week 40 − weeks 30–33)/weeks 30–33) ×100) value to demonstrate changing temporal trends in gravid populations from mid-season to end-season timepoints.

**Reporting summary**. Further information on research design is available in the Nature Portfolio Reporting Summary linked to this article.

## Data availability

Source data for Figs. 1–4 are available in Supplementary Data 1. Mosquito surveillance datasets analyzed during the current study are available from the authors and their respective organizations on reasonable request.

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

## Acknowledgements

We would like to thank Julie Coughlin of the Iowa Department of Public Health and the many local public health partners that contributed to our mosquito trapping efforts. We would like to thank Dr. Philip Dixon for assistance in our statistical analysis, Dr. James Klimavicz for coding and statistics help, and Dr. Ryan Tokarz for creating the initial map of *Culex* species distributions. Data from California were provided through CalSurv data request #48 (https://vectorsurv.org). This research was supported by the USDA National Institute of Food and Agriculture, Hatch Project 101071, the Epidemiology and Laboratory Capacity for Infectious Diseases (ELC) Program through the Iowa Department of Public Health, and the Midwest Center of Excellence for Vector-Borne Disease. This publication was supported by Cooperative Agreement #U01 CK000505, funded by the Centers for Disease Control and Prevention. CMB acknowledges support from the Pacific Southwest Center of Excellence in Vector-Borne Diseases funded by the U.S. Centers for Disease Control and Prevention (#1U01CK000516). Its contents are solely the responsibility of the authors and do not necessarily represent the official views of the Centers for Disease Control and Prevention or the Department of Health and Human Services.

## Author contributions

E.N.F. and R.C.S. designed the research, E.N.F. and R.C.S. performed the research, analysis, and data visualization, J.J.S., M.C.E., K.J.P., B.J.W., K.J., B.B., C.A., G.D.E., P.M.A., C.M.B., and R.C.S. contributed surveillance data, E.N.F. and R.C.S. wrote the initial draft of the manuscript, with all authors contributing to revisions of the final manuscript.

## Competing interests

The authors declare no competing interests.
