## [Peer Review File · Communications Biology]

Reviewers' comments:

Reviewer #1 (Remarks to the Author):

Field et al. examined the natural diapause timeline for *Culex pipiens* across the United States, by combining data from laboratory, semi-field, and surveillance studies. They found that diapause induction starts in late August, and is shaped by temperature and photoperiod, as well as additional factors such as latitude, elevation, and mosquito genetics. This work is original and furthers the field as most previous studies on *Culex pipiens* diapause are relying on laboratory experiments. The study is well executed and methods are particularly powerful due to the combination of laboratory, semi-field, and surveillance data. Data visualization is excellent and it was a true pleasure to read this well-written manuscript. Overall, this work provides important insights in the diapause induction of the main mosquito vector for West Nile virus, which has implications for virus transmission and control. I only have a few minor comments, which are listed below:

1. Lines 177-179: How did the different conditions affect survival of larvae used in the laboratory and semi-field experiments? If this data was recorded, it would be helpful to include it in the manuscript to show how rearing conditions affected survival.
2. Methods: It would be helpful if the methods would be described in a bit more detail. Particularly, how many replicates were included in the laboratory and semi-field experiments (within each year)?
3. Lines 527-529: Can you justify the use of one-way ANOVA for the laboratory comparisons (e.g. did data follow a normal distribution)?

Reviewer #2 (Remarks to the Author):

The manuscript reports the importance of photoperiod and temperature on diapause induction and provided temporal evidence of the seasonal conditions that naturally promote diapause in central Iowa, USA. Authors employed gravid *Cx. pipiens* population data from Iowa and several other places across the United States to act as a proxy for diapause incidence to put these data in the perspective of diapause occurrence in wild mosquito populations. These findings imply that the natural diapause ecology is highly influenced by temperature, latitude, elevation, and *Culex* population genetics. In sum, these findings shed important new light on the intricate process of *Cx. pipiens* diapause induction and how it affects patterns in the end-of-season mosquito population. These findings have significant health implications for the spread of diseases caused by mosquitoes, and the authors deepen our understanding of how a changing environment may affect mosquito overwintering and extend mosquito activity.

To improve the manuscript quality, some suggestions are listed below.

1. Fig 2 c and d. to confirm the diapause phenotype, the author examines differences in ovarian follicle morphology and fat body lipid staining with Nile Red. These are convincing results but there should be bars in the figures comparing the relative sizes of follicle/fat body cells between diapause and non-diapausing inducing conditions.

2. In the results of Figures 2c and d, the number of caught mosquitoes decreased during the 38–39-week period, and thus the number of samples decreased accordingly. This resulted in a natural decrease in mosquito populations as the temperature decreased. However, due to the small sample size, it is difficult to properly estimate the average size of the follicles at this time. Therefore, caution should be exercised in assessing the induction rate of diapause with the average value of the follicle size due to the decrease in the number of mosquitoes caught during the 38-39 weeks. Also, to date, no detailed study has been conducted on how many females in the mosquito population that did not start the diapause program in late summer or early autumn stop blood-feeding and ovarian

development for other reasons. Therefore, more attention should be paid to calculating the diapause induction rate in the field conditions by comparing simple light-trap and gravid-trap. Despite these shortcomings, if terms such as confirmed are avoided in the conclusion and discussion sections, the results of figure 2 and figure 3 have important value as the first diapause study using field data.

3. Although it is a study that relies on theoretical modeling, it is a recent study closely related to the current study, so if the author compares the results of this paper with the results of this paper in the course of the discussion, I think it will become a more interesting topic for discussion.

"Arora, A.K., Sim, C., Severson, D.W. and Kang, D.S., 2022. Random Forest Analysis of Impact of Abiotic Factors on *Culex pipiens* and *Culex quinquefasciatus* Occurrence. *Frontiers in Ecology and Evolution*, 9, p.773360."

Response to Reviewer's comments

COMMSBIO-22-2114

"Semi-field and surveillance data define the natural diapause timeline for *Culex pipiens* across the United States"

A detailed response to each of the reviewer comments is listed below. All changes in response to the reviewer's comments are highlighted with the "tracking changes" function in the manuscript text.

Reviewer #1

- 1. Lines 177-179: How did the different conditions affect survival of larvae used in the laboratory and semi-field experiments? If this data was recorded, it would be helpful to include it in the manuscript to show how rearing conditions affected survival.**

We would like to thank the reviewer for their comment. Unfortunately, we did not take detailed measurements of lab survival in either the laboratory or semi-field experiments. While we did record start/end dates of development for each semi-field cohort (Figure S3), we did not measure the number of mosquitoes that had progressed from 1st instar to adults in these conditions. As a result, the decreased survival (at least in these semi-field) conditions were observations, and not directly measured. As a result, we cannot make any firm assertions of the percentage of mosquitoes that had survived each condition. We have added new text in our revised manuscript (lines 186-187) to make this more transparent.

- 2. Methods: It would be helpful if the methods would be described in a bit more detail. Particularly, how many replicates were included in the laboratory and semi-field experiments (within each year)?**

We would like to thank the reviewer for their comment. In the revised manuscripts, we have added additional experimental details regarding replicates in the laboratory (lines 434-435) and semi-field experiments (lines 465-468 and 480-481).

- 3. Lines 527-529: Can you justify the use of one-way ANOVA for the laboratory comparisons (e.g. did data follow a normal distribution)?**

We would like to thank the reviewer for catching this. The mosquito measurements (ovaries and wing length) should have been analyzed using a non-parametric test (Kruskal-Wallis) since they do not have a normal distribution. These results have been reanalyzed for statistical significance using this new analysis in our revised manuscript. In addition, we have added additional text in the legend for Figure 1 and the methods section (lines 537-540) to address these changes.

Reviewer #2

- 1. Fig 1 c and d. to confirm the diapause phenotype, the author examines differences in ovarian follicle morphology and fat body lipid staining with Nile Red. These are convincing results but there should be bars in the figures comparing the relative sizes of follicle/fat body cells between diapause and non-diapausing inducing conditions.**

We would like to thank the reviewer and apologize for the oversight. We have added scale bars to the micrographs in Figure 1C and 1D in our revised manuscript. We have also included text in the figure legend to comment on the size of the scale bars.

- 2. In the results of Figures 2c and d, the number of caught mosquitoes decreased during the 38–39-week period, and thus the number of samples decreased accordingly. This resulted in a natural decrease in mosquito populations as the temperature decreased. However, due to the small sample size, it is difficult to properly estimate the average size of the follicles at this time. Therefore, caution should be exercised in assessing the induction rate of diapause with the average value of the follicle size due to the decrease in the number of mosquitoes caught during the 38-39 weeks. Also, to date, no detailed study has been conducted on how many females in the mosquito population that did not start the diapause program in late summer or early autumn stop blood-feeding and ovarian development for other reasons. Therefore, more attention should be paid to calculating the diapause induction rate in the field conditions by comparing simple light-trap and gravid-trap. Despite these shortcomings, if terms such as confirmed are avoided in the conclusion and discussion sections, the results of figure 2 and figure 3 have important value as the first diapause study using field data.**

We would like to thank the reviewer for their comment. We agree with the reviewer that the low sample size in our late-season semi-field cohorts make any statistical comparisons difficult. This is one reason why we calculated diapause incidence as the percentage of resulting mosquitoes with follicles $\leq 50\mu\text{m}$ of the total, not the average follicle size as the reviewer mentions above.

We also agree with the reviewer that there is very little information regarding the seasonal impacts on mosquito physiology, and the potential that temperature (and other abiotic factors) can also influence blood-feeding, fecundity, etc. as the reviewer suggests. This is supported by the impacts of cooler temperatures on blood-feeding (Figure 2F) and on body size (Figure 2E) provided in our study, yet further work is needed to further explore potential non-diapause physiological impacts on mosquito biology.

However, it is unclear what the reviewer means by “more attention should be paid to calculating the diapause induction rate in the field conditions by comparing simple light-trap and gravid-trap.” When using surveillance data, we are unable to calculate a diapause incidence rate. The gravid trap data used in our study only serve as a proxy for diapause induction, where decreases in gravid trap abundance reflect reductions in reproductive mosquito populations. We infer that this corresponds with diapause incidence based on the timing of these annual declines and correlations to our semi-field studies, but this cannot be directly stated, nor do we attempt to. As opposed to a diapause incidence rate, we instead measure population declines over the approximate period of week 30-40 that strongly correlate with conditions that promote *Cx. pipiens* diapause.

- 3. Although it is a study that relies on theoretical modeling, it is a recent study closely related to the current study, so if the author compares the results of this paper with the results of this paper in the course of the discussion, I think it will become a more interesting topic for discussion. "Arora, A.K., Sim, C., Severson, D.W. and Kang, D.S., 2022. Random Forest Analysis of Impact of Abiotic Factors on *Culex pipiens* and *Culex quinquefasciatus* Occurrence. *Frontiers in Ecology and Evolution*, 9, p.773360."**

We would like to thank the reviewer for the suggestion. We have incorporated this reference into the discussion of our revised manuscript (lines 339-342 and 373-376).

REVIEWERS' COMMENTS:

Reviewer #2 (Remarks to the Author):

The authors addressed the key issues well. I recommend for the publication.